# Analysis of Medical Services for Insomnia in Korea: A Retrospective, Cross-Sectional Study Using the Health Insurance Review and Assessment Claims Data

**DOI:** 10.3390/healthcare10010007

**Published:** 2021-12-22

**Authors:** Chaewon Son, Yu-Cheol Lim, Ye-Seul Lee, Jung-Hwa Lim, Bo-Kyung Kim, In-Hyuk Ha

**Affiliations:** 1Jaseng Hospital of Korean Medicine, 536 Gangnam-daero, Gangnam-gu, Seoul 06110, Korea; sohn.sppcf@gmail.com; 2Jaseng Spine and Joint Research Institute, Jaseng Medical Foundation, 3F, 538 Gangnam-daero, Gangnam-gu, Seoul 06110, Korea; hmh6692@gmail.com (Y.-C.L.); yeseul.j.lee@gmail.com (Y.-S.L.); 3Department of Neuropsychiatry, School of Korean Medicine, Pusan National University, Gumo-ro 20, Mulgeum-eup, Yangsan 50612, Korea; suede22@hanmail.net; 4Department of Neuropsychiatry, Pusan National University Korean Medicine Hospital, Gumo-ro 20, Mulgeum-eup, Yangsan 50612, Korea

**Keywords:** insomnia, medical service utilization, cost of care, hypnotics, sedatives

## Abstract

This study aimed to analyze current trends in healthcare utilization and medication usage in patients with insomnia. We reviewed the National Patient Sample data from the Health Insurance Review and Assessment Service to determine healthcare utilization in patients diagnosed with insomnia (International Classification of Diseases-10 codes G470, F510) between January 2010 and December 2016. There were 87,470 patients enrolled in this study who utilized healthcare services at least once during the 7-year period. Healthcare utilization trends, Korean and Western medicine (KM and WM, respectively) therapies utilized, comorbidities, and socioeconomic data were analyzed. The number of patients seeking WM or KM care for insomnia increased annually. Adults aged ≥45 years accounted for 73% of the cohort, and there were more female than male patients. KM treatment including acupuncture was the most common in KM (65.29%), while examination was the most common WM treatments (49.31%). In pharmacological therapy, sedatives and hypnotics were the most common (41.08%), followed by antianxiety (19.50%), digestive system and metabolism-related drugs (7.77%). The most common comorbidities were mental health disorders (50.56%) in WM but musculoskeletal disorders in KM (35.67%). Code G470 was used more frequently than code F510, and the difference was more evident in KM than in WM. The findings will provide valuable information for both clinicians and researchers.

## 1. Introduction

Insomnia is defined as the persistence of symptoms such as difficulty initiating or maintaining sleep or early-morning waking at least three times a week for ≥3 months [1]. Insomnia is a very common disorder, and intermittent short-term insomnia affects approximately 30–50% of the total population. A study in the United States (US) and United Kingdom (UK) reported an acute insomnia prevalence of 9.5% and 7.9%, respectively, with an annual incidence of 31.2–36.6% [2]. A study in Korea reported that one in five adults develops insomnia [3]. The prevalence of insomnia also increases with age; approximately 50% of older adults have difficulty initiating or maintaining sleep [4]. Further, the prevalence of insomnia according to the International Classification of Sleep Disorders-2 (ICSD-2) criteria was 32.7% among Korean older adults [5].

Insomnia is multifactorial [6]. Smoking, alcohol use, and diminished physical activity are associated with insomnia in older adults. The risk factors for insomnia include being divorced, separated, or widowed (for women); having a low education level; and having a low income level [7]. Insomnia may cause severe pain and injury in the body, as well as fatigue, daytime sleepiness, cognitive impairment, and mood disorders [6]. Sleep deprivation can result in deterioration of the overall quality of life, characterized by depression and poor work performance, and chronic insomnia can increase mortality by activating the inflammatory processes in the body and inducing cardiovascular diseases [8,9]. 

The goal of insomnia treatment is to improve sleep and reduce any resultant suffering or functional impairments [10]. Cognitive behavioral therapy for insomnia (CBT-I) is suggested as the first-line treatment for adult chronic insomnia by the American College of Physicians clinical practice guidelines (2016), European Sleep Research Society clinical practice guidelines (2017), and American Academy of Sleep Medicine guidelines for the assessment and treatment of adult chronic insomnia (2008) [10,11,12]. In general, CBT-I comprises: (1) education regarding normal sleep, sleep hygiene, and the purpose of CBTI-I, (2) stimulus control therapy, (3) sleep restriction therapy, (4) relaxation techniques, and 5) cognitive therapy [13]. If CBT-I alone is inadequate or ineffective, pharmacological therapy can be considered. Appropriate doses of drugs for treating adult sleep initiation and sleep maintenance difficulties are used separately, and pharmacological therapy beyond four weeks is not recommended. Drugs used in Korea for sleep initiation include zolpidem, triazolam, and ramelteon; those used for sleep maintenance or early morning awakenings include eszopiclone, doxepin, trazodone, suvorexant, and zolpidem controlled-release (CR) [14].

The degree of symptom improvement achieved with insomnia treatment differs among patients. Previous studies report an insomnia persistence rate of 40–69% over a period of 1–20 years [15], suggesting that chronic insomnia can be a substantial financial burden on society [16]. According to a World Health Organization report, insomnia is ranked eleventh on the list of mental, neurological and substance-use disorders with the greatest disease burden worldwide [17]. A 2010 European study comparing the direct and indirect costs of various brain diseases ranked insomnia as the ninth among neuropsychiatric disorders [18]. Although specific estimates can vary widely depending on the methodology, insomnia-related direct and indirect cost estimates in the US are reported to be 2–16 billion USD and 75–100 billion USD, respectively [16]. The indirect costs were mostly incurred by worker absenteeism, presenteeism (diminished daytime productivity), and occupational accidents [19]. A Korean epidemiology study on sleep disorders reported that 22.8% of 5000 adults had insomnia [20], and a Health Insurance Review and Assessment Service (HIRA) report showed that the number of patients being treated for insomnia increased from 405,000 in 2015 to 633,000 in 2019, with the total amount of covered health benefits increasing from 38.7 billion KRW in 2012 to 66.7 billion KRW in 2016 [21].

The escalating prevalence of insomnia, and consequent direct and indirect costs of care, demand an examination of the latest trends in insomnia care. As previously mentioned, Western clinical practice guidelines recommend CBT-I followed by pharmacological therapy to treat insomnia. However, there are limitations in the clinical implementation of CBT-I, and medication compliance is low among patients because of concerns regarding developing tolerance and dependence and adverse drug reactions with prolonged use, such as disturbance of sleep flow [22]. As a result, alternative therapy is sometimes used to treat insomnia; popular alternatives include phototherapy, exercise therapy, and acupuncture therapy [10,23,24]. Research on such alternative therapies is ongoing. One study investigating the short-term effects of acupuncture on sleep quality reported that acupuncture improved sleep efficiency and total sleep duration compared with placebo treatment [25]. Acupuncture therapy is less costly compared with psychotherapies, is not time consuming, and can be adjusted depending on the targeted symptom [26]. 

South Korea features a bimodal healthcare system, wherein patients with insomnia have the option to receive covered care at Western medicine (WM) or Korean medicine (KM) facilities. KM treatments include acupuncture, moxibustion, cupping therapy, herbal medicine, and KM psychotherapy. Korean patients can receive WM and KM treatments concomitantly; therefore, there may be differences in the limitations of these treatments and their socioeconomic burden in Korea compared with that from other countries, which highlights the need for studies that consider both KM and WM treatments. Previous studies primarily examined the treatment trends in WM, with a focus on analyzing the relationship between insomnia and a single factor such as sex, age, and type of pharmacological therapy [5,27,28].

This study aimed to investigate the characteristics of insomnia patients, available WM and KM services, utilization of these services, medical costs, and comorbidities and analyze their trends using the 2010 to 2016 Health Insurance Review and Assessment (HIRA) claims data. The HIRA-National Patient Sample (NPS) data are nationally representative data that enable a comparison of WM and KM utilization by patients with insomnia. We ultimately aimed to present reliable foundational data for clinicians, researchers, and policymakers to contribute to effective healthcare policies.

## 2. Materials and Methods

### 2.1. Data Source

We used the HIRA-NPS data from January 2010 to December 2016, employing data in which the neuropsychological disease codes remained unmasked. The HIRA data are insurance claims data generated when healthcare providers submit their claims for reimbursement of costs of healthcare provided to patients. Because >98% of the Korean population is covered by national health insurance, these data are nationally representative and, thus, highly valuable for healthcare research [29]. These data provide detailed information on various factors: details of care (e.g., treatment, procedure, test, drug prescriptions), diagnosis, cost covered by the insurer, patient’s out-of-pocket cost, patient’s demographic information, and information on the healthcare facility.

The HIRA-NPS includes claims data of randomly selected patients (3%; approximately 1,375,842 as of 2011) from the total population and is stratified by sex (two strata) and age (16 strata). Since only a small percentage of the patients are included every year, the dataset is analyzed under the assumption that there is a very low possibility of overlapping patients. These are secondary data sampled from the raw data after removing information regarding individuals and legal entities; the data contain annual claims with reference to the first day of care of the corresponding year.

### 2.2. Study Design and Study Population

The International Classification of Diseases-Tenth Revision (ICD-10) codes used for diagnosing insomnia are F codes (psychiatrics) and G codes (neurology), and the frequently used codes include F510 (nonorganic insomnia), F518 (other nonorganic sleep disorders), F519 (nonorganic sleep disorder, unspecified), F519A (emotional sleep disorder NOS), G470 (disorders of initiating and maintaining sleep, insomnias), G470A (chronic insomnia), G470B (acute insomnia), G472 (disorders of the sleep-wake schedule), G472C (irregular sleep-wake patterns), G478 (other sleep disorders), and G479 (sleep disorder, unspecified). In this study, only patients who received WM or KM care at least once with the code F510 (nonorganic insomnia) or G470 (disorders of initiating and maintaining sleep, insomnias) as the main diagnosis between 2010 and 2016 were included. These criteria were adopted from a previous Korean study [28].

Among the claims submitted with Korean Standard Classification of Diseases (KCD) codes F510 or G470, cases with a code for dentistry or public health facility (6167 cases); cases with the type of institution listed as a dental hospital, maternity clinic, or public health facility (225 cases); cases of patients diagnosed with a cancer-related code during the observation period (76,262 cases); or cases with the total cost and number of days in care entered as 0 or missing (1448 cases) were excluded.

### 2.3. Study Outcomes

General medical service use of insomnia patients according to the type of visit was investigated by each year. The patients whose data were selected for the analysis were classified based on baseline characteristics such as age, sex, payer type, type of visit, and medical institution, and the frequency and percentage of each parameter. Age was divided into eight categories of 10-yearly units from <15 years to ≥75 years; payer type was divided into health insurance, Medicaid, and others. The type of visit was classified into outpatient care and inpatient care based on the claim, and medical institution was divided into tertiary hospital/general hospital/hospital, clinic, KM hospital, and KM clinic.

The service category in the HIRA-NPS database was divided into examination fee, injection fee, KM treatment, psychotherapy fee, hospitalization, radiological diagnosis, test, treatment/surgery, and others, and the frequency and total expense for each category were analyzed. We analyzed cases of acupuncture (acupoint, special acupoint, zone, and device), a treatment in which needles are injected into acupoints based on KM pattern identification, and these cases were specified as KM treatment. Total expense was defined as the sum of the expenses of care paid by the patient’s out-of-pocket costs and insurer, which is covered by the National Health Insurance Service. 

The service codes for insomnia-related treatment were divided into WM and KM for analysis. The number of services, total expenses, number of patients, average annual expense per service, and average annual expenses per patient were analyzed for psychotherapies in WM and for acupuncture, moxibustion, cupping therapy, and psychotherapies in KM. Psychotherapies in KM were categorized into four different therapies (a. Gyeongja-pyeongji therapy, b. Oji-sangseung therapy, c. Ijeong-byeongi therapy, and d. Jieon-goron therapy) [30,31]. The main characteristic of the psychotherapy in KM is its theoretical basis, more specifically theories in Korean Medicine in understanding emotions and psychological disorders, which can be explained in two parts: one is that emotions and psychological states are interrelated, which allows moderation or enhancement of one emotion by another; two is that emotions can be resolved by not only conversations and interactions but also by external stimuli such as physical stimulations [32].

We categorized the inpatient and outpatient drug prescriptions according to the anatomical therapeutic chemical classification codes with reference to the Ministry of Health and Welfare classification criteria and analyzed the frequency and expenses for each category. Prepared herbal medicine was excluded as it is not covered by health insurance. Comorbidities of insomnia were analyzed separately for WM and KM for all main and sub-diagnosis codes, including F510 and G470. Among the codes for insomnia, codes equivalent to ICD codes F510 and G470 were analyzed separately for WM and KM to compute the total number and percentage of claims with each of these codes.

### 2.4. Statistical Analysis

We used univariate linear regression model to evaluate if the annual trend in general medical use were statistically significant in each category. The basic characteristics of the patients are presented as the number of patients and percent. The frequency and expenses of each service category and prescription category were analyzed. Each visit were defined as inpatient or outpatient treatments, and categorized by the types of medical institutions. Number of cases were further analyzed by different subtypes of insomnia disease codes of F510 and G470. Types of interventions were defined to analyze the frequency and total expenses. Comorbidities of each patient were categorized by the 1st level and the distribution of comorbidities were shown in terms of number of patients and percentages. All prescriptions were categorized by ATC codes. General medical services were presented by the number of patients, number of cases, total expenses, annual expenses, and annual visits per patient per year. The average log change (as growth rate) of each category were investigated. All cost values were converted to USD based on the 2020 average KRW to USD exchange rate and adjusted using the consumer price index for the healthcare and medical sector of the corresponding year. The data were analyzed using SAS software (version 9.4, SAS Institute, Cary, NC, USA).

## 3. Results

### 3.1. Healthcare Costs

During the period 2010–2016, a total of 426,232 claims were submitted with ICD-10 codes F510 or G470 for insomnia as the main diagnosis. Among these, a total of 339,492 claims for 87,470 patients were selected for analysis in this study (Figure 1).

As shown in Table 1, a total of 9881 patients sought healthcare for insomnia in 2010, with 1303 visiting a KM facility and 8578 visiting a WM facility; in contrast, a total of 15,362 patients sought healthcare for insomnia in 2016, with 1826 visiting a KM facility and 13,536 visiting a WM facility. This finding indicates that the number of patients seeking medical care for insomnia increased annually over the 7-year study period (*p*-value < 0.0001). While the total number of cases seeking WM care tended to increase over the years, the number of patients seeking KM care did not increase markedly (Figure 2, Appendix A). The total expense of care in WM consistently increased every year, while that in KM increased only slightly (Appendix A). The average per-patient expense of care in KM remained similar over the 7-year period, while that of WM increased every year except for 2011. The average expense per case generally increased consistently over the years for WM, but the annual changes were not consistent for KM, with a higher average expense of care in KM than in WM (Appendix A). The number of visits to healthcare facilities remained steady over the study period, and the number of visits to a KM facility was approximately two-fold higher than that to a WM facility (Appendix A). 

### 3.2. Baseline Characteristics and Healthcare Utilization

As shown in Table 2, more female patients (62.38%) than male patients (37.62%) sought healthcare for insomnia. Most patients were middle aged or older, with the highest number of patients aged 55–64 years (19.60%), followed by those aged 45–54 years (19.40%), 65–74 years (18.74%), and ≥75 years (15.25%). Most patients only sought WM care (n = 76,780, 88%), while 8979 patients utilized only KM and 1711 utilized both WM and KM. The age distribution was similar between patients utilizing only KM (55–64 years, 21.97%; 45–54 years, 20.84%; 65–74 years, 18.90%) and those utilizing only WM (55–64 years, 19.20%; 45–54 years, 19.19%; 65–74 years, 18.59%). The main age distribution of patients who utilized both KM and WM was as follows: 55–64 years (25.07%); 65–74 years (24.49%); and 45–54 years (21.57%). The percentages of male and female patients were 38.96% and 61.04%, respectively, among patients only utilizing WM and 27.36% and 72.64%, respectively, among patients only utilizing KM, indicating that a higher percentage of female patients sought KM care than WM care. The percentages of male and female patients seeking both KM and WM care were 31.39% and 68.61%, respectively, also demonstrating that a higher percentage of women sought care. Health insurance was the predominant payer type (92.95%), followed by Medicaid (6.93%). Appendix A shows the frequency and percentage of the utilization of outpatient/inpatient care and level of healthcare facility for KM and WM. 

### 3.3. Number of Medical Cases and Medical Costs per Category 

As shown in Table 3, examination was the most commonly practiced category (531,353 services, 45.33%), followed by medication administration (287,899 services, 24.56%), KM treatment (147,832 services, 12.61%), and psychotherapy (82,426 services, 7.03%). In WM, examination was the most commonly practiced category (466,399 services, 49.31%), followed by medication administration (276,114 services, 29.19%) and psychotherapy (82,426 services, 8.72%). In KM, KM treatment was the most commonly practiced category (147,832 services, 65.29%), followed by examination (64,954 services, 28.69%). 

The category that incurred the greatest total expense over the 7-year period was examination (3,508,362 USD, 52.84%), followed by psychotherapy (1,216,909 USD, 18.33%) and medication administration (702,345 USD, 10.58%). In WM, the categories with the highest total expense were examination (3,082,432 USD, 54.61%), psychotherapy (1,216,909 USD, 21.56%), and medication administration (684,515 USD, 12.13%), while in KM, the categories with the highest total expense were KM treatment (523,799 USD, 52.62%) and examination (425,930 USD, 42.79%).

### 3.4. Frequently Prescribed Medications for Insomnia

As shown in Table 4, the most frequently prescribed drugs were sedatives and hypnotics (208,524 cases), followed by antianxiety drugs (99,000 cases) and antidepressants (68,145 cases). The total expense was the highest for sedatives and hypnotics (640,755 USD), followed by antidepressants (376,083 USD) and digestive and metabolism-related drugs (186,367 USD). The per-case expense (6.97 USD) and per-patient expense (38.96 USD) were the highest for antipsychotics.

### 3.5. Specific Interventions for Insomnia in WM and KM

We analyzed frequently performed treatments for insomnia in WM and KM, except for examination, medication administration, and physical therapy (Appendix A). The total expense for psychotherapy in WM was 1,216,909 USD (82,426 cases). According to treatment category, the total expense was the highest for supportive therapy (786,797.12 USD, 63,478 cases), followed by intensive therapy (323,320.61 USD, 13,870 cases), intensive analytic therapy, and individual treatment. In KM, the total expense was the highest for acupuncture (408,779.61 USD, 103,001 cases), cupping (50,319.11 USD, 11,254 cases), moxibustion (44,772.76 USD, 16,544 cases), and KM psychotherapy (3,758.12 USD, 202 cases). Among KM psychotherapies, gyeongja-pyeongji treatment was most frequently prescribed (1,184.12 USD, 42 cases), followed by oji-sangseung treatment, ijeong-byeongi treatment, and jieon-goron treatment. The per-patient expense for psychotherapy was higher in KM than in WM. Furthermore, psychotherapy incurred the highest annual expense per patient for insomnia patients in both WM (61.35 USD) and KM (43.20 USD) (data not shown). 

### 3.6. Numbers of Cases of Insomnia with Comorbidities

As shown in Table 5, we analyzed the comorbidities of insomnia separately for WM and KM. In terms of the ICD-10 code blocks, the top five blocks for comorbidities were F codes (mental and behavioral disorders) (n = 77,064), K codes (disease of the digestive system) (n = 16,858), M codes (diseases of the musculoskeletal system and connective tissue) (n = 12,727), J codes (diseases of the respiratory system) (n = 11,342), and I codes (diseases of the circulatory system) (n = 6737) in WM. In KM, the top five blocks were M codes (n = 8925), F codes (n = 3591), K codes (n = 3288), U codes (codes for special purposes) (n = 2591), and R codes (symptoms, signs, and abnormal clinical and laboratory findings) (n = 2095). Mental disorders (F codes), musculoskeletal disorders (M codes), and digestive diseases (K codes) were included in the top three comorbidity code blocks in both WM and KM. The results were consistent even when WM and KM were analyzed together (F codes, n = 80,655; M codes, n = 21,652; K codes, n = 20,146), and the next most common code blocks were J codes (n = 12,086) and R codes (n = 7957). U codes for KM diagnosis and diseases were common in KM, while I codes for circulatory diseases were common in WM.

### 3.7. Number of Cases According to the ICD-10 Codes F510 and G470

The number of claims submitted for insomnia care provided at healthcare facilities was 15,509 containing code F510 and 20,195 containing code G470 in 2010 and 26,083 containing code F510 and 36,218 containing code G470 in 2016, showing that claims for both codes increased. Over the 7-year period, code G470 was used more frequently than code F510, and this difference was more evident in KM than in WM. In WM, code F510 was used more frequently than code G470 with the exception of 2010 and 2016, and in KM, code G470 was used more frequently than code F510 (Figure 3).

## 4. Discussion

### 4.1. Healthcare Utilization and Medication Usage

There were approximately 1.7-fold more female patients than male patients among those who sought healthcare for insomnia, which is consistent with previous findings that insomnia more frequently affects women than men [8]. Most patients who sought healthcare for insomnia were in the 55–64 years, 45–54 years, 65–74 years, or ≥75 years age groups, with 73% of the patients aged ≥45 years. There was a high percentage of women aged 45–75 years among patients who sought healthcare for insomnia. The age range of patients seeking KM care for insomnia was 35–74 years, which was similar to the age range of patients seeking KM care for other diseases [33].

Among patients who sought healthcare for insomnia, children and adolescents aged < 15 years accounted for the smallest proportion (0.41%). Although sleep disorders are quite common among children and adolescents, with a prevalence of approximately 27–62% [34], their healthcare utilization may be the lowest because only approximately 50% of caregivers consult a physician for their children’s sleep disorders [35]. Compared to other age groups, children and adolescents aged < 15 years preferred KM care (3.04%) to WM care (0.11%). This contrasts with the preference for WM care (16.13%) over KM care (8.15%) among adults aged ≥75 years. As children and adolescents aged < 15 years have distinct characteristics in terms of healthcare utilization, such as being accompanied by their caregivers when seeking healthcare, further research is needed in this group.

Regarding inpatient and outpatient care utilization as determined based on claims data, outpatient care utilization was remarkably higher than inpatient care at 99.82%. Outpatient care utilization at primary health facilities was high in both KM and WM, with 70.77% utilizing a WM clinic and 15.84% utilizing a KM clinic. Insomnia is a very common health problem, and sleep problems frequently occur alongside existing psychological and physical problems. Thus, the prevalence can be high (above 50%) among patients who visit primary care facilities that are easily accessible [36].

Insomnia is often accompanied by other physical and psychological symptoms [37]. Poor sleep quality and shortened sleep duration may exacerbate symptoms of depression and can contribute to chronic pain by aggravating central sensitization [38]. Previous studies reported that 53% of patients with chronic pain seek healthcare for sleep-related symptoms [39]. Many studies have provided evidence that irritable bowel syndrome, gastro-esophageal reflux disease, and functional dyspepsia are associated with sleep disorders; the likelihood of developing additional sleep dysfunctions and comorbidities such as anxiety and depression increases with increasing severity of dyspepsia [40]. In our analysis of comorbidities of insomnia, mental disorders (F codes), musculoskeletal disorders (M codes), and digestive diseases (K codes) ranked in the top three comorbidities in both WM and KM. In WM, a high percentage of patients suffered from mental health issues. In KM, a high percentage of patients had a musculoskeletal disorder. According to the 2017 survey of KM care utilization, musculoskeletal and connective tissue-related conditions were the most common reasons (61.1%) for seeking outpatient KM care, followed by digestive conditions (10.9%) [33]. This result supports previous findings that acupuncture is effective in treating chronic pain associated with musculoskeletal disorders [41].

In a study of patients with insomnia who visited the Sleep Clinic at the National Center for Mental Health in 2018, Soh (2019) reported that these patients had been treated at a different healthcare facility for sleep problems before visiting the Sleep Clinic and that pharmacological treatment (92%) was used in most cases [42]. In light of our finding that medication administration was the second-highest category following examination, we can infer that pharmacological treatments are frequently used to treat insomnia in WM. The analysis of frequently prescribed drugs revealed that sedatives and hypnotics were most frequently prescribed, followed by antianxiety drugs and digestive and metabolism-related drugs. Such prescription trends seem to be linked to the top-ranked comorbidities in WM, namely other anxiety disorder (F41), depressive episode (F32), and gastritis and duodenitis (K29). 

The most commonly used sedatives and hypnotics were zolpidem, triazolam, and flunitrazepam. Triazolam is a benzodiazepine (BZD) that extends sleep duration and shortens sleep latency; however, withdrawal of the drug may trigger a relapse of insomnia, more severe rebound insomnia, and withdrawal symptoms. Furthermore, triazolam has a short half-life and, thus, may cause more intense withdrawal symptoms. Zolpidem is a BZD receptor agonist that has recently become popular owing to its comparable effects to those of BZDs with reduced tolerance and habituation. Despite the fact that it is effective as a short-term treatment (≤4 weeks) for early- and mid-stage insomnia, the use of zolpidem requires caution as it is associated with similar adverse events to those of BZDs (e.g., cognitive decline and falls) and may cause neuropsychiatric adverse events (e.g., parasomnias, amnesia, hallucinations) [43,44,45]. Thus, hypnotics may be associated with an array of adverse events; in addition, the use of hypnotics is associated with high cancer incidence and mortality [46,47]. 

There are several challenges in the clinical implementation of CBT-I, the standard non-pharmacological therapy for insomnia, as it is costly and time consuming, and approximately 20% of patients with insomnia are nonresponsive to treatment, with 39–44% failing to retain the therapeutic effects after treatment [21]. We could not examine CBT-I in this study because its health insurance coverage did not commence until 2018. Thus, additional studies are required to investigate the trends and cost of CBT-I. 

According to a study conducted among KM doctors, the most common reason for visiting a KM facility among insomnia patients was to enhance their sleep quality and reduce the use of hypnotics, and the most frequently performed treatment was acupuncture along with the administration of herbal medicine [48]. Among various KM psychotherapies, our results showed that ijeong-byeongi therapy and oji-sangseung therapy were most frequently performed (86 cases and 31 cases, respectively). Acupuncture, the most widely performed treatment for insomnia in KM, is associated with few adverse events but high efficacy in the nervous and endocrine systems, and it has been substantiated as an effective alternative medicine modality in many clinical trials [49]. Acupuncture is less costly and can be performed more quickly than psychotherapies. Furthermore, while WM drugs target fixed and specific symptoms of insomnia, acupuncture has an added benefit of adjusting acupoints depending on the targeted symptom [26]. 

Our analysis of the frequency of insomnia codes F510 and G470 revealed that code G470 was more frequently used than code F510 during the study period, and the difference was more evident in KM than in WM. This is consistent with previous findings that KM doctors frequently use code G470 for insomnia [48]. It is possible that they preferred a G code over an F code because of the patients’ psychological burden associated with an F code and the fact that most patients with insomnia also have other symptoms. In 2013, there was an attempt to resolve the negative views and social stigmatization of mental disorders by amending the law to change the main diagnosis to “counseling.” Bias not only leads to discrimination against people with mental disorders in their daily lives but also contributes to building a stereotype and negative image of mental disorders. Private and public sectors may intentionally or unintentionally limit opportunities for individuals with mental disorders [50]. A 2016 survey on mental disorders reported that the rate of mental health service utilization by the Korean population remained markedly low at 9.6%, compared to 43% among the American population (2015) [51]. Hence, changing the perception of mental illnesses and improving mental health services for the affected individuals would be a significant step.

Insomnia often presents as a comorbidity of several other diseases; Thus, it is important to administer individualized treatments for patients by accurately identifying patterns of insomnia, such as difficulty with sleep initiation, maintenance or early morning awakening, as well as each patient’s living environment and medical history [52]. While sleep hygiene education and CBT are recommended as first-line treatments for insomnia, it is difficult to implement these interventions in Korea owing to the relatively high cost, lack of practitioners with expertise, and current medical fee system. Therefore, the numbers of patients seeking healthcare for insomnia and hypnotic prescriptions will inevitably increase. The higher utilization of WM than KM observed in our study is presumably because of drug prescriptions. Hypnotics should be prescribed with caution not only due to the associated adverse reactions but also owing to the issues of rebound insomnia, withdrawal symptoms, and recurrence that occur when medications are reduced or stopped [52]. Performing electroacupuncture while to taper off sleeping pills was effective in reducing adverse reactions that typically occur with prolonged use beyond four weeks [53]. Pragmatic observational studies are required to identify effective concomitant treatments and alternatives to pharmacological treatment that are clinically confirmed to improve sleep.

### 4.2. Strengths and Limitations

There are several strengths to this study. Firstly, the data used in this study were collected from the entire population of the country and, thus, are nationally representative. Secondly, we observed a long-term period of 7 years, from 2010 to 2016, the latest available data for analysis. Thirdly, this is the first study to analyze trends in insomnia care in both KM and WM in Korea. We analyzed the main treatment modalities and their cost, as well as the frequency of prescribed treatments and medications to treat insomnia, in each specialty. 

This study had a few limitations. First, among the cases with insomnia as the main diagnosis, we only analyzed those with the code G470 or F510 with reference to the criteria used in previous studies. Hence, patients who were treated for insomnia with other codes as the main diagnosis or sub-diagnosis may have been omitted. However, because insomnia is associated with various symptoms, we included cases in which insomnia was used as a sub-diagnosis because of differences in the severity of the symptoms in our analysis of comorbidities. Second, we could not analyze clinical manifestations of insomnia (e.g., difficulty of sleep initiation or maintenance) because we used claims data. Furthermore, the degree of impact of insomnia on patients’ lives was not examined using scales such as the Insomnia Severity Index or Pittsburgh Sleep Quality Index. Additional studies are needed to investigate varying trends in healthcare utilization according to clinical manifestations and severity of insomnia. Third, it is possible that non-covered treatments such as herbal medicine and chuna therapy were omitted. Moreover, we could not compute the frequency and cost of CBT-I in this study because insurance coverage of CBT-I, which is specifically performed at the neuropsychiatry clinics in WM, was not available during the study period. Subsequent studies should also examine non-covered categories. Fourth, the study included repeated cross-sectional data that enable the analysis of follow-up care over a period of 1 year, but the data lacked yearly continuity. Cohort studies are needed for the long-term analysis of follow-up care. Finally, we analyzed sample data but could not perform an in-depth analysis of psychotherapies performed in KM facilities owing to the smaller number of psychotherapies performed relative to that in WM facilities. This is attributable to the smaller number of KM psychiatrists and the narrower scope of sleep disorder-related psychotherapies that can be performed by KM neuropsychiatrists, thus calling for relevant policy measures. 

## 5. Conclusions

In this study, we presented the general trends in the treatment of insomnia in Korea, including the types of treatments, cost of care, and comorbidities, by analyzing the HIRA-NPS data. The number of patients visiting a primary care facility for insomnia increased annually along with the cost of care, implying an increasing importance of diagnosing insomnia in the clinical field. Furthermore, the overall utilization of pharmacological and non-pharmacological interventions is analyzed to reflect recent practices in Korea. Although psychotherapy is recommended for insomnia prior to prescribing drugs such as hypnotics and sedatives, there are several challenges, such as cost and clinical implementation. Thus, we highlight the need to complement psychotherapies with a focus on clinical utilization, to devise criteria to prevent prolonged use of drugs, and to develop measures to alleviate adverse reactions. In addition, KM utilization for treating insomnia is analyzed and discussed, which warrant further studies regarding its clinical effectiveness. The findings of this study will be useful as a foundation for further investigation of standard insomnia treatments and costs of care.

## Figures and Tables

**Figure 1 healthcare-10-00007-f001:**
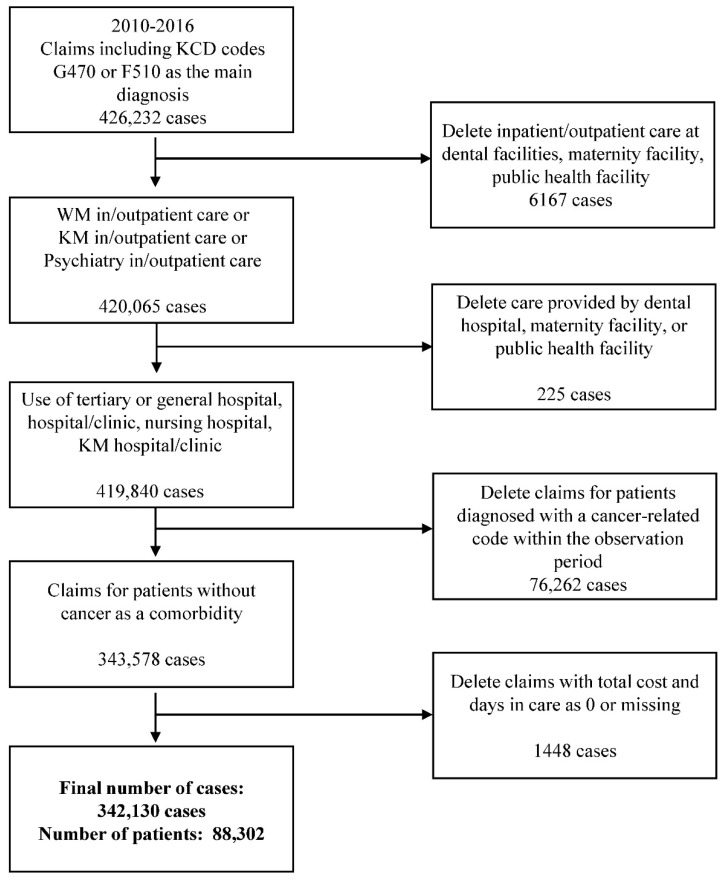
Flowchart of the study sample. KCD = Korean Standard Classification of Diseases; WM = Western medicine; KM = Korean medicine.

**Figure 2 healthcare-10-00007-f002:**
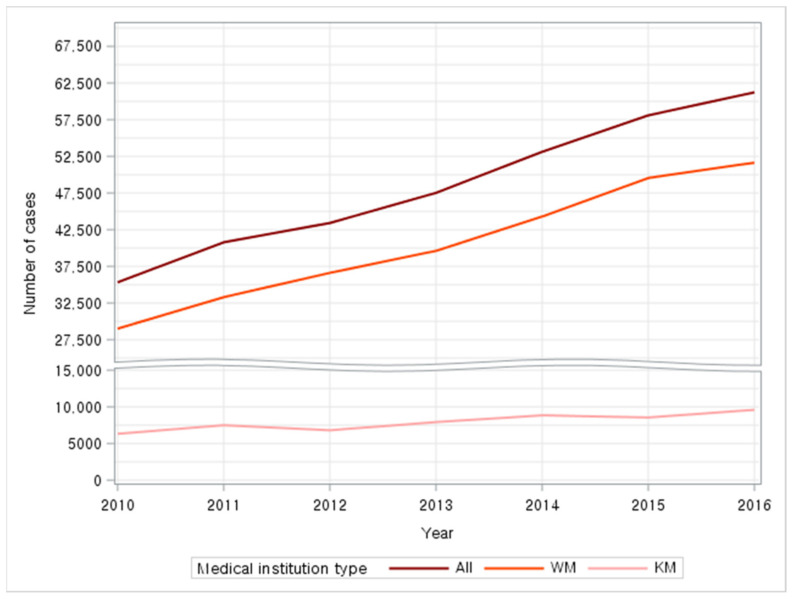
General medical service use for insomnia—number of cases; WM = Western medicine; KM = Korean medicine.

**Figure 3 healthcare-10-00007-f003:**
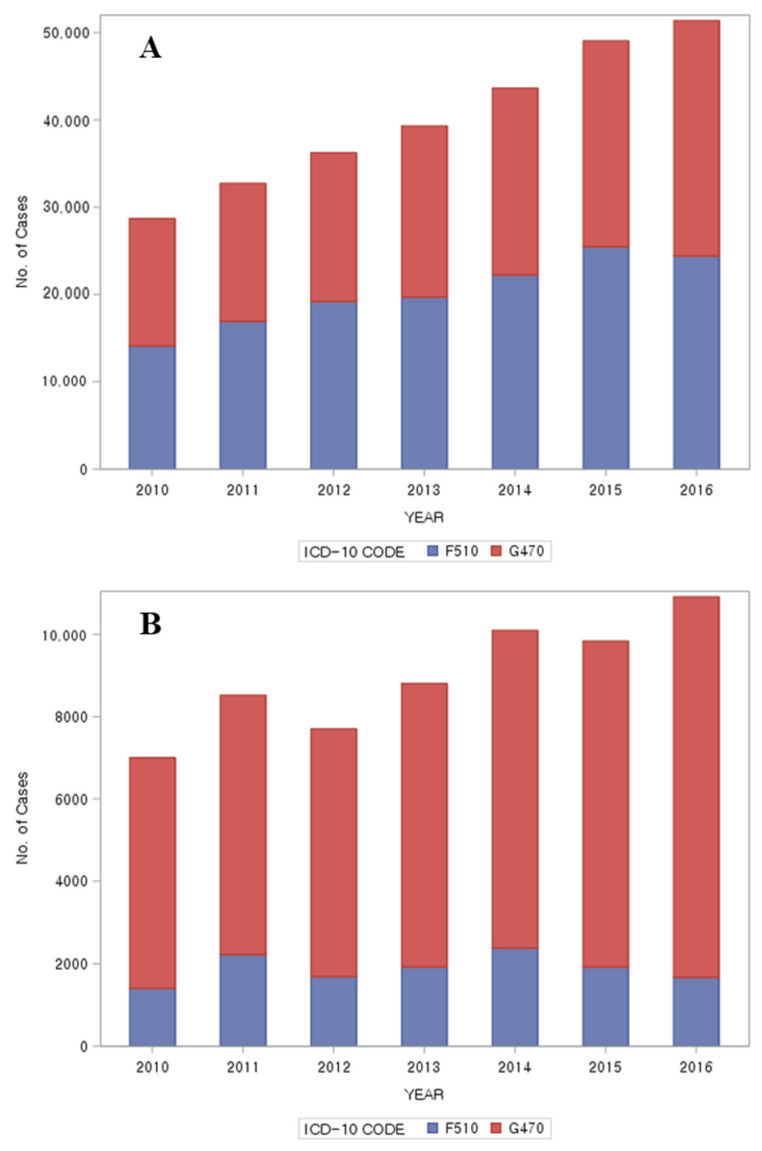
Number of cases per ICD-10 code F510, G470; (**A**): Western medicine; (**B**): Korean medicine. ICD-10 = International Classification of Diseases, Tenth Revision.

**Table 1 healthcare-10-00007-t001:** General medical service use for patients with insomnia.

Category	Type of Visit	Year							*p*-Value *
		2010	2011	2012	2013	2014	2015	2016	
No. of patients	Total	9881	11,303	11,649	12,471	13,618	14,897	15,362	<0.0001
	WM	8578	9858	10,303	11,020	11,997	13,199	13,536	<0.0001
	KM	1303	1445	1346	1451	1621	1698	1826	0.0016
No. of cases	Total	35,328	40,778	43,413	47,523	53,124	58,091	61,235	<0.0001
	WM	29,002	33,292	36,608	39,611	44,287	49,546	51,642	<0.0001
	KM	6326	7486	6805	7912	8837	8545	9593	0.0022
Total expense	Total	$529,941	$699,089	$728,091	$890,754	$1,032,866	$1,129,685	$1,143,220	<0.0001
	WM	$442,862	$586,277	$626,568	$761,495	$873,214	$979,806	$978,872	<0.0001
	KM	$87,079	$112,812	$101,523	$129,259	$159,652	$149,879	$164,348	0.0019
Per-patient	Total	$53.63	$61.85	$62.50	$71.43	$75.85	$75.83	$74.42	0.0030
	WM	$51.63	$59.47	$60.81	$69.10	$72.79	$74.23	$72.32	0.0021
	KM	$66.83	$78.07	$75.43	$89.08	$98.49	$88.27	$90.00	0.0269
Per-case expense	Total	$15.00	$17.14	$16.77	$18.74	$19.44	$19.45	$18.67	0.0140
	WM	$15.27	$17.61	$17.12	$19.22	$19.72	$19.78	$18.95	0.0190
	KM	$13.77	$15.07	$14.92	$16.34	$18.07	$17.54	$17.13	0.0076
Total care	Total	141,791	188,944	210,626	230,259	267,531	297,266	286,923	0.0002
	WM	134,732	180,738	202,962	221,534	257,775	287,662	276,514	0.0003
	KM	7059	8206	7664	8725	9756	9604	10,409	0.0010
Av. days in care per patient	Total	24.15	29.38	29.09	29.73	32.78	33.51	30.13	0.0478
	WM	15.88	18.51	19.84	20.25	21.66	21.93	20.58	0.0163
	KM	8.27	10.87	9.25	9.48	11.13	11.58	9.55	0.2982
Total visits	Total	37,613	42,724	43,949	48,875	53,951	59,770	62,077	<0.0001
	WM	31,070	35,067	37,114	40,883	45,027	51,111	52,452	<0.0001
	KM	6543	7657	6835	7992	8924	8659	9625	0.0034
Avr visits per patient	Total	9.35	9.78	9.48	9.94	10.20	9.83	10.00	0.0666
	WM	3.77	3.72	3.74	3.83	3.91	4.00	4.02	0.0024
	KM	5.58	6.06	5.74	6.10	6.29	5.84	5.98	0.3436

* Univariate linear regression model was used to estimate the annual trend of each category. Abbreviations: KM: Korean Medicine; WM: Western Medicine.

**Table 2 healthcare-10-00007-t002:** Basic characteristics of patients.

		Total		Western Medicine		Korean Medicine		Both Medicines	
		No. of Patients	%	No. of Patients	%	No. of Patients	%	No. of Patients	%
Age	<15 years	359	0.41	85	0.11	273	3.04	1	0.06
	15–24	3108	3.55	2614	3.40	468	5.21	26	1.52
	25–34	8566	9.79	7647	9.96	825	9.19	94	5.49
	35–44	11,596	13.26	10,301	13.42	1140	12.70	155	9.06
	45–54	16,971	19.40	14,731	19.19	1871	20.84	369	21.57
	55–64	17,142	19.60	14,740	19.20	1973	21.97	429	25.07
	65–74	16,392	18.74	14,276	18.59	1697	18.90	419	24.49
	≥75 years	13,336	15.25	12,386	16.13	732	8.15	218	12.74
Sex	Male	32,909	37.62	29,915	38.96	2457	27.36	537	31.39
	Female	54,561	62.38	46,865	61.04	6522	72.64	1174	68.61
Payer type *	NHI	81,302	92.95	71,080	92.58	8605	95.83	1617	94.51
	Medicaid	6064	6.93	5597	7.29	374	4.17	93	5.44
	Others	104	0.12	103	0.13	-	-	1	0.06

* NHI: National Health Insurance; WM = Western medicine; KM = Korean Medicine.

**Table 3 healthcare-10-00007-t003:** Number of medical services and medical expenses per category.

Service Category	Total				Western Medicine				Korean Medicine			
	No. of Services	%	Expenses	%	No. of Services	%	Expenses	%	No. of Services	%	Expenses	%
Examination	531,353	45.33	$3,508,362	52.84	466,399	49.31	$3,082,432	54.61	64,954	28.69	$425,930	42.79
Medication administration	287,899	24.56	$702,345	10.58	276,114	29.19	$684,515	12.13	11,785	5.20	$17,830	1.79
KM treatment	147,832	12.61	$523,799	7.89	-	-	-	-	147,832	65.29	$523,799	52.62
Psychotherapy	82,426	7.03	$1,216,909	18.33	82,426	8.72	$1,216,909	21.56	-	-	-	-
Test	59,587	5.08	$248,335	3.74	59,587	6.30	$248,335	4.40	-	-	-	-
Injection	39,101	3.34	$52,208	0.79	39,101	4.13	$52,208	0.93	-	-	-	-
Treatment/surgery	15,959	1.36	$48,490	0.73	15,959	1.69	$48,490	0.86	-	-	-	-
Diagnostic radiology	3965	0.34	$39,915	0.60	2715	0.29	$34,137	0.60	1250	0.55	$5778	0.58
Hospitalization	2874	0.25	$280,292	4.22	2638	0.28	$259,955	4.61	236	0.10	$20,337	2.04
Other *	1205	0.10	$18,662	0.28	843	0.09	$16,987	0.30	362	0.16	$1675	0.17

* Other: anesthesia, 100 co-insurance; all cost-related results presented in this study were converted to the 2020 level based on healthcare and medical service price index adjusted for healthcare inflation rate and KRW:USD exchange rate (see Appendix A). KM = Korean medicine.

**Table 4 healthcare-10-00007-t004:** Medications used for patients with insomnia.

Category	No. of Prescriptions	Total Expenses	No. of Patients	Expenses per Prescription	Annual Expenses per Patient
Sedatives and hypnotics	208,524	$585,696	62,189	$2.81	$9.42
Antianxiety drugs	99,000	$146,077	25,732	$1.48	$5.68
Antidepressants	68,145	$345,930	14,530	$5.08	$23.81
Digestive and metabolic drugs	39,418	$170,716	15,483	$4.33	$11.03
Musculoskeletal	23,626	$53,415	11,236	$2.26	$4.75
Antipsychotics	13,740	$88,522	2486	$6.44	$35.61
Anticonvulsants	12,101	$11,832	2667	$0.98	$4.44
Cardiovascular	10,282	$39,639	3132	$3.86	$12.66
Respiratory	9497	$15,616	5412	$1.64	$2.89
Antihistamines	8429	$18,744	4560	$2.22	$4.11
Other	14,878	$99,959	7300	$6.72	$13.69

All cost-related results presented in this study were converted to the 2020 level based on the healthcare and medical service price index, adjusted for healthcare inflation rate and KRW:USD exchange rate (see Appendix A).

**Table 5 healthcare-10-00007-t005:** Comorbidities of patients with insomnia.

Western Medicine			Korean Medicine		
Diagnostic Code *	No. of Patients	%	Diagnostic Code *	No. of Patients	%
F	77,064	50.56%	M	8925	35.67%
K	16,858	11.06%	F	3591	14.35%
M	12,727	8.35%	K	3288	13.14%
J	11,342	7.44%	U	2591	10.35%
I	6737	4.42%	R	2095	8.37%
R	5862	3.85%	G	1150	4.60%
G	5202	3.41%	S	763	3.05%
E	4587	3.01%	J	744	2.97%
L	3229	2.12%	H	517	2.07%
N	3035	1.99%	N	450	1.80%
H	2610	1.71%	E	390	1.56%
B	1143	0.75%	I	236	0.94%
S	1107	0.73%	L	202	0.81%
A	493	0.32%	B	32	0.13%
Z	242	0.16%	Z	24	0.10%
T	93	0.06%	A	12	0.05%
Q	58	0.04%	T	11	0.04%
Y	16	0.01%	O	1	0.00%
O	1	0.00%	-	-	-

* A, B: certain infectious and parasitic diseases; E: endocrine, nutritional and metabolic diseases; F: mental, behavioral and neurodevelopmental disorders; G: diseases of the nervous system; H: diseases of the eye, adnexa, ear, mastoid; I: diseases of the circulatory system; J: diseases of the respiratory system; K: diseases of the digestive system; L: diseases of the skin and subcutaneous tissue; M: diseases of the musculoskeletal system and connective tissue; N: diseases of the genitourinary system; O: pregnancy, childbirth and the puerperium; P: certain conditions originating in the perinatal period; Q: congenital malformations, deformations and chromosomal abnormalities; R: symptoms, signs and abnormal clinical and laboratory findings, not elsewhere classified; S, T: injury, poisoning and certain other consequences of external causes; U: codes for special purposes; V, Y: external causes of morbidity; Z: factors influencing health status and contact with health services.

## Data Availability

Patient Samples can be obtained via website of HIRA by completing the End User Agreement of the Patient Samples. The Patient Samples are provided in a DVD (text file) format, and a fee is charged for the samples. https://opendata.hira.or.kr/home.do (accessed on 10 December 2021).

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
