# Peer review of "Analysis of Medical Services for Insomnia in Korea: A Retrospective, Cross-Sectional Study Using the Health Insurance Review and Assessment Claims Data"

_healthcare, 2021, doi:10.3390/healthcare10010007_

Round 1

Reviewer 1 Report

Manuscript ID: healthcare-1462469

Title: Analysis of medical services for insomnia in Korea: A retrospective, cross-sectional study using the Health Insurance Review and Assessment claims data

Journal: Healthcare

Abstract

  1. Authors could add some information about the sample being examined. For example, they could specify the overall number of patients being considered in this retrospective study.

Introduction

  1. Lines 63-64. Authors should also specify the drugs being used for the early morning awakening.
  2. Line 70. I am not convinced that insomnia is a brain disease.

Materials and Methods

  1. Line 114. Authors should clarify why they chose to examine this time interval.
  2. Authors should move the Tables to the Results section.

Results

  1. Lines 286-288. Authors should describe, within the Materials and Methods section, these treatments.

Discussion

  1. Line 329. Which are the main differences between WM and KM psychotherapies?
  2. Lines 412-413. Regardless of the duration of the single session, it would be extremely important to know the mean number of sessions per each therapy.
  3. Lines 416-418. Can a therapy last for less than 15 minutes? I am not convinced.
  4. Line 450. It seems to me that the early morning awakening is completely neglected.
  5. Line 464. Authors should separately present strengths and limitations.
  6. Lines 473-474. Authors wrote “Hence, patients who were treated for insomnia with other codes as the main diagnosis or sub-diagnosis may have been omitted”. Why?
  7. Lines 479-480. A scale cannot be objective by definition.

Reviewer 2 Report

Thank-you for the opportunity to review this paper. Overall, I think this is a well-written, useful paper but some aspects of the statistical analysis could be clearer. 

Introduction:

Line 37: Did the authors of the paper referenced (reference 2) report 95% confidence intervals for these estimates of prevalence and incidence? If so, I would include here.

Line 72: Do you have a reference for these estimates of direct and indirect costs in the US?

The authors could justify why only data up till 2016 was analysed, considering this was 5 years. I suspect the reasons are practical but I would like to see the justification.

Methods:

I find it strange that the authors refer to results tables in the Methods section. I would place the results tables in the Results section and only refer to them at that point. The authors should be able to explain their methods without referring to results tables.

Line 150: Did the authors consider that there may be a non-linear trend of cases over time?

Table 1: The table includes p-values but it does not say anywhere what test this p-value is from and what exactly was tested. This could be included in a footnote.

Table 2: Again, it is not explicitly said what the p-values reported are testing. Is the chi-square test comparing the characteristics across western, Korean and both? This could be clarified in a footnote.

Line 201: “Chi-square test was used to test for significant differences in the demographic character between groups”. By groups, do the authors mean KM and WM? This should be clarified.

Results:

Line 222 – what test is this a p-value from? Is it the linear trend of patients over time? The authors should be explicit here that the test was for a linear trend specifically.

Table 2 and baseline characteristics: Presumably these are the baseline characteristics totalled from over the 2010-2016? If so, please clarify. Do the authors recognise that potentially many people put in claims over multiple years, hence there may be some overlap?

Line 242: I don’t understand what this sentence is saying. Are the age groups 55-64, 65-74 and 45-54 the most common age categories for those who accessed both KM and WM?

Line 244: Although the difference in age distribution across the three groups was statistically significant, if you look at the percentages, they are mostly fairly similar across the three groups. Because of the large sample size, even differences that are not clinically meaningful may be statistically significant. The authors could be more aware of this.

Table 5: Could some patients not have more than one comorbidity? If so, the categories in Table 5 would not be discrete and the chi-square test would not be appropriate.

Supplementary material: The Figures of trends over time are a nice way of seeing the data. I would consider putting at least one of these figures in the main manuscript.

Reviewer 3 Report

Thank you for the opportunity to review this interesting manuscript. I have a few minor comments for the authors to consider:

  • Abstract:
    • Lines 22-30: Please include the actual results, e.g. the number, the %.
  • Introduction:
    • Lines 105-107: The aims stated by the authors in this section covered more aspects than that mentioned in the abstract (i.e. service utilization only). It is fine to be more explicit about th study aims in the introduction but it would be good for the authors to state their primary aim (e.g. service utilization) and their secondary aims (e.g. patient characteristics, services available, medical costs, etc). The authors may also state their research hypotheses instead. The hypothesis statements would justify the report of p-values in the tables and results section (descriptive analyses does not require report of p-values).
  • Methods:
    • Section 2.3: Please only list the study outcomes in this section. Delete the mention of statistical methods used and reference of any results tables.
    • Section 2.4:
      • line 196: Linear regression is not the correct method for count outcome variables (i.e. number of service use). The authors could use the Chi-square test or (nominal) logistic regression.
      • Line 201-202: Unless the authors wanted to test the association between demographic characteristics and groups (please state so in the introduction section), there is no need to perform chi-squared tests. Similarly for comorbidities (line 205).
    • Results: Depending on the research hypotheses, the authors may remove report of p-values from some tables.
    • Discussion:
      • Lines 318-331: These sentences are not the discussion of any results. Please delete.
      • Lines 403-419: These sentences are not the discussion of any results. Please delete.
      • Section 4.2: This section is well written and honestly reflected the strengths and limitations of the current study.
      • The authors may add a last paragraph outlining the (clinical) implications of the reported findings.

Round 2

Reviewer 1 Report

Manuscript ID: healthcare-1462469

Title: Analysis of medical services for insomnia in Korea: A retrospective, cross-sectional study using the Health Insurance Review and Assessment claims data

Journal: Healthcare

Authors have adequately addressed my previous concerns. I do not have any further suggestions.

This manuscript is a resubmission of an earlier submission. The following is a list of the peer review reports and author responses from that submission.